# Obesity-Induced TNFα and IL-6 Signaling: The Missing Link between Obesity and Inflammation—Driven Liver and Colorectal Cancers

**DOI:** 10.3390/cancers11010024

**Published:** 2018-12-27

**Authors:** Lara Kern, Melanie J. Mittenbühler, Anna Juliane Vesting, Anna Lena Ostermann, Claudia Maria Wunderlich, F. Thomas Wunderlich

**Affiliations:** Max Planck Institute for Metabolism Research, Cologne Excellence Cluster on Cellular Stress Responses in Aging-associated Diseases (CECAD), Center for Endocrinology, Diabetes and Preventive Medicine (CEDP), 50931 Cologne, Germany; lara.kern@sf.mpg.de (L.K.); melanie.mittenbuehler@sf.mpg.de (M.J.M.); Anna.Vesting@sf.mpg.de (A.J.V.); anna.ostermann@sf.mpg.de; (A.L.O.); claudia.wunderlich@sf.mpg.de (C.M.W.)

**Keywords:** Obesity, low-grade inflammation, liver and colon cancer, IL-6 and TNF, signaling

## Abstract

Obesity promotes the development of numerous cancers, such as liver and colorectal cancers, which is at least partly due to obesity-induced, chronic, low-grade inflammation. In particular, the recruitment and activation of immune cell subsets in the white adipose tissue systemically increase proinflammatory cytokines, such as tumor necrosis factor α (TNFα) and interleukin-6 (IL-6). These proinflammatory cytokines not only impair insulin action in metabolic tissues, but also favor cancer development. Here, we review the current state of knowledge on how obesity affects inflammatory TNFα and IL-6 signaling in hepatocellular carcinoma and colorectal cancers.

## 1. Obesity-Induced Inflammation Impairs Insulin Action

More than two decades ago, Hotamisligil and colleagues discovered the negative impact of tumor necrosis factor α (TNFα) on insulin sensitivity in obesity [1,2]. Since then, numerous research articles have been published, reviewed elsewhere (e.g., [3]), that support the view of obesity-induced, low-grade inflammation in the development of insulin resistance. 

In obesity, the enlargement of the white adipose tissue (WAT) induces mechanical as well as endoplasmatic reticulum (ER) stress in adipocytes, leading to the release of free fatty acids (FFA) and inflammatory cytokines (Figure 1) [1,4]. Subsequently, recruitment of immune cells to the obese WAT enhances local and systemic inflammation [5]. This obesity-driven, low-grade, chronic inflammation affects insulin sensitivity of all metabolic organs such as WAT, liver, muscle, pancreas, and even the central nervous system [6]. 

The obese WAT differs from that of lean individuals regarding immune cell composition and numbers. In particular, cell types of innate and adaptive immunity have been shown to infiltrate the obese WAT (Figure 1) [7,8]. One major subset of infiltrating innate immune cells are macrophages that, depending on their polarization, can exert variable functions in WAT [9,10]. In detail, Mantovani et al., defined polarized macrophages in two classes: M1 proinflammatory and M2 anti-inflammatory macrophages [11]. 

In the obese WAT, proinflammatory M1 macrophages form crown-like structures around dead adipocytes and thereby contribute to the obesity-induced, low-grade inflammation [10,12]. The elevated number of M1 macrophages in the obese WAT is the main source of TNFα and interleukin-6 (IL-6) (Figure 1) [4]. In contrast, lean WAT contains more M2 macrophages that mediate anti-inflammatory functions [9]. At the front between innate and adaptive immunity, natural killer cells (NK) are activated and proliferate in the obese WAT to trigger M1 macrophage accumulation and polarization [13]. Furthermore, obesity promotes the expansion of a distinct IL-6 receptor α (IL-6Rα) positive NK cell subpopulation, which is also present in the blood of obese patients, expressing myeloid lineage markers to contribute to the obesity-induced inflammation [14]. Besides innate immune cells, adaptive immunity mediated by T and B cells are also involved in WAT inflammation. In line, cluster of differentiation (CD)4 and CD8-positive effector T cells both contribute to the obesity-induced inflammation [15,16]. While in lean WAT, regulatory T cells (Treg) are anti-inflammatory, the percentage of these cells is reduced in the obese WAT [17]. Although not fully understood, B cells were reported to control Treg numbers in the WAT. Obese B-cell-deficient mice have increased WAT-resident Tregs and, consequently, a reduction in inflammation as well as improved insulin sensitivity [18]. Thus, both innate and adaptive immune cells contribute to obesity-induced inflammation and the development of insulin resistance. 

However, other cytokines, such as IL-1, monocyte chemoattractant protein 1 (MCP1), IL-10, and IL-17, are also altered upon obesity and have been extensively examined [5,19,20,21]. IL-6 signaling is not only required to induce IL-10 expression, but is also involved in T cell differentiation to IL-17-expressing T cells, thereby further complicating the exact definition of cell type specific contributions of inflammatory mediators in obesity and its associated disorders [22,23]. 

The contribution of TNFα and IL-6 signaling to obesity-associated insulin resistance is well studied in conditional mouse models, but still reveals contradictory outcomes. TNFα the IR- substrate 1 (IRS1) required for downstream activation of insulin signaling [2]. On the one hand, TNFα activates spingomyelinase to produce ceramides that convert IRS1 into an inhibitor of the intrinsic tyrosine kinase of IR [24,25]. On the other hand, TNFα activates intracellular c-JUN N-terminal kinase (JNK) 1 and IκB kinase (IKK) that blunt insulin signaling via inhibitory phosphorylation of IRS1 at serine 307, which is a read-out of TNFα’s negative impact on insulin sensitivity (Figure 2) [26,27]. Consistently, JNK1 but not JNK2 deficiency retained insulin sensitivity in diet-induced obesity [28]. However, exchanging S307 in IRS1 to a nonphosphorylatable amino acid fails to protect against the development of obesity-induced insulin resistance, and instead impairs insulin sensitivity, indicating that the impact of TNFα on insulin signaling might be context- and cell-type-dependent [29]. 

In contrast to TNFα, IL-6 exerts both pro- and anti-inflammatory effects. IL-6-mediated STAT3 activation leads to the expression of the negative regulator of IL-6 signaling, suppressor of cytokine signaling (SOCS), which in turn destabilizes IRS1, thereby blunting insulin action (Figure 2) [30]. While hepatic SOCS3 deficiency initially prevents impaired insulin action in diet-induced obesity, these mice have accelerated inflammation at later stages [31]. This suggests a fine-tuned balance between anti- and proinflammatory actions of IL-6. Contradictory to its role as a proinflammatory cytokine, inactivation of classical membrane-bound IL-6Rα signaling in hepatocytes and macrophages shows impaired insulin sensitivity [32,33]. Otherwise, mice with IL-6Rα deficiency in T cells exhibit improved insulin sensitivity at the onset of diet-induced obesity [34]. Taken together, TNFα and IL-6 signaling in obesity-associated inflammation point towards complex, temporal, and cell-type-specific functions in the development of insulin resistance. 

However, TNFα and IL-6 not only impact systemic insulin sensitivity in obesity, but also promote cancer, since these cytokines are also produced from cells of the tumor microenvironment [35]. In line, TNFα and IL-6 levels are elevated in the serum of cancer patients. In fact, most cancer types show a positive correlation between cancer incidence and body mass index [36,37,38]. While multiple factors contribute to obesity-associated cancers, we will only focus on the role of obesity-induced TNFα and IL-6 signaling in the promotion and development of hepatocellular carcinoma (HCC) and colorectal cancer (CRC) within this review.

Insulin binding to the insulin receptor (IR) results in the activation of the intrinsic tyrosine kinase of IR and subsequent tyrosine phosphorylation of the IRS1, ultimately causing protein kinase B (AKT) activation. Obesity increases systemic levels of tumor necrosis factor α (TNFα) and interleukin-6 (IL-6). TNFα signaling activates intracellular c-JUN N terminal kinase (JNK) and IκB kinase (IKK). Both blunt insulin signaling by inhibitory phosphorylation of IRS1 at serine 307. TNFα signaling triggers the production of ceramides through the sphingomyelin phosphodiesterase (SM). Ceramides suppress the phosphorylation of IR, thereby inhibiting the insulin signaling pathway. IL-6 binds to the IL-6 receptor α (IL-6Rα), which recruits two glycoprotein 130 (GP130) receptor molecules to activate janus kinase (JAK). JAK phosphorylates tyrosine residues of the GP130 receptor and thereby induces activation of signal transducer and activator of transcription 3 (STAT3). Phosphorylated STAT3 translocates into the nucleus and induces target gene expression. In turn, suppressor of cytokine signaling 3 (SOCS3) is expressed and inhibits the IL-6 signaling pathway and ubiquitinates IRS1, resulting in its degradation.

## 2. TNFα and Its Bivalent Role in Liver Homeostasis 

TNFα is one of the master regulators of liver homeostasis. It mediates hepatocyte proliferation, but also controls hepatocyte apoptosis and necroptosis [39,40]. Secreted TNFα binds either TNF receptor 1 (TNFR1) or TNF receptor 2 (TNFR2) [41]. However, the main signal transduction occurs via TNFR1. TNFR activates proinflammatory nuclear factor κB (NF-κB) and mitogen-activated protein (MAP) kinase pathways rapidly through signaling complex I [42], which is composed of the adaptor molecule TNF type 1-associated DEATH domain protein (TRADD), the kinase receptor interacting serine/threonine-protein kinase 1 (RIP1), as well as TNFR-associated factor 2 (TRAF2). The NF-κB heterodimer composed of p65 and p50 is sequestered in the cytosol by inhibitor of κBα (IκBα), when cells are not exposed to TNFα. TNFα-mediated activation of the IKK complex, which is comprised of the two catalytic subunits, IKK1 and IKK2, as well as the regulatory subunit NF-κB essential modulator (NEMO), induces classical NF-κB pathway activation through IKK-mediated phosphorylation and degradation of IκBα (Figure 2). Degradation of IκBα leads to nuclear translocation of the p65/p50 heterodimer and subsequently the induction of NF-κB-dependent target gene expression such as proinflammatory and cell survival genes [43]. NF-κB induces transcription of antiapoptotic target genes such as cellular inhibitors of apoptosis (cIAPs), cIAP1, cIAP2, X-linked inhibitor of apoptosis protein (XIAP), and B cell lymphoma 2 (Bcl2) family members [44,45]. 

In addition to IKK complex/NF-κB signaling, the JNK pathway has emerged as one of the key regulators of hepatic inflammation (Figure 2) [46]. Here, TNFR activation leads to the formation of complex I and activation of transforming growth factor *β*-activated kinase 1 (TAK1) [46]. TAK1 further activates mitogen-activated protein kinase kinase 4 (MKK4) and MKK7, which subsequently leads to JNK activation and downstream cJUN activation.

## 3. TNFα: A Key Driver of Hepatocellular Carcinoma (HCC) Development

The obesity-induced, nonresolving, inflammatory microenvironment is a major driver of tumor progression, especially of HCC, characterized by the presence of immune cells and proinflammatory cytokines such as TNFα [47,48]. The main source of TNFα is liver-resident Kupffer cells and macrophages, which are anatomically in close proximity to hepatocytes [42]. A key step in tumor initiation is compensatory proliferation of hepatocytes. Recruited macrophages secrete TNFα and IL-6 to block hepatocyte apoptosis and in turn induce hepatocyte survival and proliferation. Permanent proliferating hepatocytes accumulate DNA damage and proliferation-induced mutations, which promote HCC development [49]. Consistently, genetic mouse models revealed that inactivation of TNFR1 decreases the incidence of liver tumors [50]. TNFR1-knockout mice fed a high fat diet show an ablation of obesity-enhanced HCC, as well as reduced obesity-induced hepatosteatosis [51]. Further, multidrug resistance protein 2 (MDR2)-deficient mice, a model of inflammation-associated HCC, develop spontaneous HCC in a TNFα-dependent manner [52]. In this model, the increased production of TNFα by endothelial and inflammatory cells activates hepatocyte-specific NF-κB signaling, thus highlighting the role of TNFα signaling in orchestrating spontaneous HCC development. Moreover, in major urinary protein (MUP)-urokinase-type plasminogen activator (uPA) transgenic mice, a model for ER and transient liver stress, blocking of TNFα and hepatic TNFR1 signaling prevents nonalcoholic steatohepatitis (NASH) in obese mice [53]. Notably, the progression of NASH to HCC is driven by TNFR1 signaling within hepatocytes, which activate protumorigenic NF-κB signaling.

During obesity-driven, low-grade inflammation, hepatic NF-κB serves as an antiapoptotic survival factor, which promotes the proliferation of HCC progenitor cells and HCC development [54]. In contrast, hepatic inactivation of IKK2 increases diethylnitrosamine (DEN)-induced HCC burden [55]. In line with this evidence, hepatic NEMO deficiency causes spontaneous progression of TNFα-mediated chronic hepatitis to HCC [56]. This detrimental effect of hepatic NEMO deficiency is potentiated under obese conditions presumably by enhanced liver inflammation and hepatic lipid accumulation. This supports the notion that NEMO acts as a tumor suppressor in the liver. Surprisingly, hepatic NEMO-deficient mice are protected against diet-induced obesity and exhibit improved insulin sensitivity [57]. A recent report demonstrated that NEMO prevents hepatocarcinogenesis via inhibiting RIP1 kinase-activity-driven hepatocyte apoptosis [58]. 

Another TNFα-induced NF-κB activator is interferon gamma (IFNγ) [59]. It was shown that TNFα and IFNγ synergistically induce the expression of ubiquitous antiapoptotic B7-H1 receptors on HCC cells via JAK/STAT/interferon regulatory factor 1 signaling, thereby enhancing adaptive immune resistance and facilitating HCC growth. [60]. Importantly, A20, also known as TNFα-induced protein 3 (TNFAIP3), a key regulator of inflammation and terminator of NF-κB signaling, reduces hepatic inflammation and cancer onset by protecting hepatocytes from TNFα-induced apoptosis under obese conditions [61,62]. Thus, hepatic NF-κB signaling might play a dual role in hepatocarcinogenesis. While in principle, NF-κB in hepatocytes is required for protection against TNFα-induced cell death, which is accelerated in obesity, the compensatory proliferation of neighboring hepatocytes promotes hepatocarcinogenesis presumably via activation of other TNFα-activated signaling pathways such as JNK.

JNKs are not only activated by TNFα, but also by FFA, which are highly abundant during overnutrition or as a consequence of excessive ceramide synthesis. Thereby, JNK induces hepatocyte lipoapoptosis by activating proapoptotic Bcl-2 proteins and triggering the mitochondrial apoptotic pathway [63,64]. Hui and colleagues [65] reported that cancer cell proliferation decreases significantly in JNK1-deficient mice and that this reduction requires JNK-dependent p21 downregulation. Interestingly, HCC progression seems to require JNK function in nonparenchymal liver cells, whereas compound deficiency of hepatocyte JNK1 and JNK2 did not prevent HCC development [66]. This finding is further emphasized by a study demonstrating that myeloid-specific JNK deficiency reduces the expression of inflammatory cytokines and hepatic infiltration of immune cells, thereby suppressing the development of HCC [67]. In line with these findings, hepatic NF-κB blockade and hepatic TAK1 deficiency lead to spontaneous progression from hepatic inflammation to liver tumors [68]. This implicates tightly controlled feedback mechanisms of both TNFα-activated pathways to prevent cancer. Taken together, TNFα-activated IKK and JNK signaling are increased in the liver upon obesity, thereby revealing their dysregulation and combined oncogenic potential in obese patients. 

TNFα and NF-κB signaling promote cancer progression by facilitating cell migration, invasion, and metastasis [45,69]. Metastasis requires extracellular matrix degradation and basement membrane destruction executed by matrix metalloproteinases (MMP), namely MMP-3 and MMP-13 [70]. TNFα increases the expression of MMP-3 and MMP-13 in HepG2 cells and enhances cancer cell migration by activation of the extracellular signal-regulated kinase/ NF-κB pathways [70]. Furthermore, reactive oxygen species (ROS) such as superoxide anion, a byproduct of respiration and a secondary messenger, have been demonstrated to drive migration. In addition, disruption of the equilibrium between ROS generation and antioxidant defense leads to oxidative stress, which induces liver DNA damage and promotes neoplastic transformations of hepatocytes [71,72,73] TNFα-induced ROS production by complex I and complex III of the mitochondrial respiratory chain activates NF-κB signaling, thereby enhancing cell migration. Metastasis is promoted by activated macrophages in the tumor microenvironment releasing TNFα and thereby driving metastasis. Finally, NF-κB is accompanied by vascular endothelial growth factor (VEGF) expression, which drives cancer progression by inducing vascularization [45]. In summary, the interplay of all these factors creates a tumor microenvironment that supports the transformation of hepatocytes, which ensures their survival and evasion from immune surveillance via antiapoptotic pathways [55,73,74].

## 4. Hepatic IL-6 Signaling: Cytokine with Versatile Functions

The pleiotropic glycoprotein IL-6 is an important signaling molecule in the activation of the immune system in response to infections and systemic inflammation [75]. IL-6 activates T cells, promotes B cell differentiation, and regulates the acute-phase response in the liver [76,77,78]. It also affects lipid metabolism, insulin resistance, and mitochondrial activity [79,80]. IL-6 further plays a role in vascular diseases, the neuroendocrine system, as well as neuropsychological behavior [81,82,83,84], hence, emphasizing the pleiotropic characteristics of this cytokine. Moreover, IL-6 was shown to maintain chronic inflammation and thus promotes disease progression, for example, autoimmune encephalomyelitis [85], arthritis [86,87], pristine-induced lupus [88], plasmacytomas [89,90,91], and various cancers including HCC [51,92,93,94,95,96,97]. 

The expression of IL-6 is induced by TNFα and IL-1 [98]. Furthermore, the Toll-like receptors, prostaglandins, adipokines, stress responses, and others are also involved in the induction of IL-6 expression [98]. The canonical IL-6 signaling pathway is transduced through the IL-6Rα (Figure 2), a transmembrane receptor found on hepatocytes, macrophages, T-lymphocytes, and endothelial cells [75]. For signal transduction, the IL-6Rα alone is not sufficient, but the ubiquitously expressed GP130 receptor molecule is necessary to induce downstream signaling [76,77,78]. Binding of IL-6 to the IL-6Rα chain recruits two GP130 chains, which activates JAK/STAT pathways [79,80]. JAK phosphorylates five tyrosine residues of GP130 and thereby induces intracellular signaling cascades including STAT3, phosphoinositide-3-kinase (PI3K)/ protein kinase B (AKT), and mitogen-activated protein kinase (MAPK) [99,100,101,102,103,104,105,106]. Besides the membrane-bound IL-6Rα, a soluble form (sIL-6Rα), which induces IL-6 trans-signaling, has been described [107]. In humans, the sIL-6Rα is either produced by alternative splicing [108] or by A disintegrin and metalloproteinase domain 10 (ADAM10)- and ADAM17-mediated shedding of the membrane-bound receptor; the latter has also been seen in mice [109]. Through association of IL-6 with sIL-6Rα, this complex can directly bind to ubiquitously expressed GP130 receptors on the cell membrane, enabling IL-6 stimulation of cells, which do not express the membrane-bound IL-6Rα [90,91].

Various cell types express the IL-6Rα, but the cell-specific effects of IL-6 signaling differ depending on the target cells. Under nonpathological conditions, deficiency of hepatic IL-6 signaling was shown to impair insulin sensitivity and glucose tolerance via accelerated TNFα expression [33]. Moreover, previous studies focusing on IL-6 signaling in immune cells attribute IL-6 an important role in the control of macrophage polarization, favoring the anti-inflammatory M2 state [32]. IL-6 signaling in NK cells was shown to drive their reprogramming into cells with myeloid gene expression and thereby impair insulin action in obesity [14]. 

## 5. IL-6 Signaling: A Fatal Player in Obesity-Induced HCC

Lack of IL-6 results in impaired immune responses after viral, parasitic, and bacterial infections, as well as impaired liver regeneration after injury [110,111,112,113,114,115,116]. Thus, IL-6 regulates, amongst others, acute phase responses and liver regeneration, resulting in dramatically increased IL-6 serum levels upon states of infection, inflammation, and liver damage [98,110,117].

IL-6 is increased in diabetes [118], obesity [119], and various cancers [51,92,93,94,95,96,97]. Elevated IL-6 serum levels strongly correlate with increased risk to develop HCC and vice versa; serum IL-6 and sIL-6Rα levels are elevated in patients suffering from HCC [120,121,122,123]. The risk of developing HCC is decreased in female mice due to increased estrogen receptor signaling, which inhibits IL-6 expression, emphasizing an important role of IL-6 signaling in HCC [124]. In consistency, whole body IL-6, IL-6Rα, and hepatic GP130 deficiency result in a significant decrease in DEN-induced HCC burden in mice [51,94,124,125]. In these knockout animals, reduced STAT3 activation is observed which results in altered gene expression of various genes involved in the regulation of proliferation, apoptosis, and fibrosis. Park and colleagues demonstrated that IL-6-deficient animals are protected from DEN-induced HCC under lean and obese conditions due to accelerated hepatic apoptosis [51]. In line with this evidence, deficiency of the IL-6Rα causes destabilization of the antiapoptotic molecule myeloid leukemia cell differentiation protein (MCL1), thereby resulting in reduced HCC burden in lean mice [94]. Specifically, IL-6 signaling inhibits the expression of molecules like protein phosphatase 1 catalytic subunit alpha (PP1CA), which is required for the activation of glycogen synthase kinase 3 beta (GSK3β). GSK3β marks MCL1 for degradation, resulting in the activation of apoptosis. Gene expression of *Mcl1 ubiquitin ligase E3* is also inhibited by STAT3. Therefore, inhibition of IL-6 signaling induces apoptosis through degradation of MCL1 [94]. However, IL-6Rα deficiency only sensitized lean, but not obese, livers to apoptosis. Therefore, obesity might induce another STAT3-activating factor that compensates for IL-6Rα deficiency. Recently, we have shown that hepatocytes from obese and from IL-6Rα-deficient mice increase their *leptin receptor* (*Lepr*) expression, implying a compensatory role of leptin signaling [126]. Ultimately, deletion of hepatic LepR in IL-6Rα-deficient mice further ameliorates DEN-induced HCC development, implicating that downstream STAT3 inhibition might be a pivotal strategy to prevent hepatocarcinogenesis in humans. In line with our results, a recent study by Grohmann and colleagues demonstrated increased hepatic pSTAT3 levels in obese mice and humans [127]. Importantly, they identified STAT3 as the driver of HCC progression in obesity, whereas obesity-driven NASH and fibrosis depend on STAT1. Taken together, these results highlight the crucial role of STAT3 signaling in obesity-associated HCC development and progression. 

In addition to classical IL-6 signaling, IL-6 trans-signaling was identified as a major driver of HCC through blocking p53-induced apoptosis and enhancing angiogenesis by inducing endothelial cell proliferation and tumor development [128]. Besides blocking apoptosis, IL-6 promotes HCC progression by upregulating osteopontin (OPN), a secretory extracellular matrix protein involved in stem cell maintenance and metastasis [129]. 

Another important factor in tumor development is angiogenesis, which ensures the required nutrient supply during tumor growth. IL-6 downstream signaling through STAT3 activates hypoxia-inducible factor 1- α (HIF-1α) and VEGF [130,131]. Niu et al., reported that tumor cells transfected with a constitutive active STAT3 mutant increases *Vegf* expression and enhances tumor angiogenesis after implantation into mice [131]. Thus, the pleiotropic nature of IL-6-induced STAT3 activation sets multiple starting points for therapeutic intervention.

IL-6 downstream signaling also involves activation of the PI3K/AKT/mammalian target of rapamycin (mTOR) signaling cascade and Villanueva and colleagues observed overactivated mTOR signaling in 50% of human HCC samples [132]. In line with this evidence, another study in human HCC linked activated mTOR signaling to poor prognosis and survival of HCC patients, suggesting an important role of PI3K/AKT/mTOR signaling in HCC progression [133]. Supporting the importance of mTOR activation during HCC progression, a previous study in mice showed that chronic activation of mTOR causes HCC [134]. mTOR indirectly activates eukaryotic translation initiation factor 4E (eIF4e), a molecule that is part of the eIF4F complex [135,136,137]. eIF4e regulates translational control of mRNAs involved in proliferation and antiapoptotic pathways and is thereby highly associated with tumorigenesis. However, whether these effects can also be linked directly to upstream IL-6 signaling remains to be elucidated. Taken together, IL-6 signaling has a pivotal role in HCC development and progression. However, given the complex downstream signaling pathways, which are activated by IL-6, some contributing factors still remain elusive. 

## 6. Obesity Disturbs the Balance of Inflammatory Effectors and Inhibitors

The activation of the IL-6 signaling pathway in hepatocytes is tightly regulated and the reaction to acute IL-6 signaling is quickly diminished due to negative feedback loops [94,138]. Consequently, deficiencies in these negative feedback loops lead to an aberrant STAT3 activation to promote oncogenic transformation [138,139,140]. Chronic inflammation in obesity disturbs the balance of inflammation effectors and their inhibitors, which might play a fundamental role in HCC development. In line with this, obese mice exhibit elevated basal activation of STAT3, but react poorly to acute IL-6 [94,126].

Excessive IL-6 signaling can be controlled through JAK/STAT degradation or inhibition by at least three different protein families. Besides SOCS proteins and src-homology 2 (SH2)-containing phosphatase (SHP-1), protein inhibitors of activated STAT (PIAS) inhibit STAT signal transduction. Chung et al., identified PIAS3 as a specific and direct inhibitor of STAT3, but not STAT1 [141]. PIAS proteins are constitutively expressed in the cell and do not terminate cytokine-induced signaling. Thus, it is assumed that PIAS3 rather fine-tunes STAT3 activation instead of serving as a classic negative feedback regulator. In contrast, SHP-1 acts in a classical negative feedback loop: upon JAK2 activation through IL-6, JAK2 phosphorylates GP130 at tyrosine residues that serve as docking sites for STAT3 [142]. However, the phosphorylation sites within GP130 are also recognized and dephosphorylated by SHP-1 to reduce the catalytic activity of JAK2 directly. In fact, SHP-1 is known to exert tumor-suppressive function in HCC [140,143]. While SHP-1 seems to be constitutively expressed in cells and instantly inhibits JAK-mediated signaling, SOCS protein expression is directly induced via STAT transcription factors and consequently its suppressive effect is delayed. This allows for time-restricted, IL-6-induced signaling. Furthermore, SHP-1 is involved in the regulation of other signaling pathways, whereas SOCS proteins act more specifically [142]. 

SOCS1-7 and cytokine-induced SH2-containing protein (CIS) are the eight members of the classic SOCS protein family [138,144]. However, IL-6 signaling induces only the expression of SOCS1 and SOCS3 [145]. Overexpression studies discovered a crucial role for SOCS3 and SOCS1 in IL-6 signaling regulation in vitro [145,146]. In contrast, Croker et al., claim that SOCS1 and SOCS3 are functionally not redundant in terms of IL-6 regulation in vivo [147]. In particular, SOCS1 regulates IFNγ activity and SOCS3 specifically controls IL-6 signaling [148]. Despite little sequence similarities of the N-terminal region, SOCS1 and SOCS3 share multiple common features. All eight members share the central SH2 domain, a conserved carboxyl-terminal region, and a conserved SOCS box [145,149]. However, a unique feature of SOCS1 and SOCS3 is the kinase inhibitor region (KIR) that acts as a noncompetitive suppressor of the JAK2 tyrosine kinase [150,151,152]. Through their SH2 domain, SOCS1 and SOCS3 mediate the direct inhibition of JAK activity or competitively block the access of STAT to phosphorylated tyrosines on the GP130 receptor [142,153]. SOCS3 binds to phosphorylated tyrosine Y757 on the receptor subunit GP130 through its SH2 domain [153]. Based on a three-dimensional model, SOCS1 is predicted to directly interact with JAK2 and inhibit its catalytic activity [153,154]. Furthermore, the SOCS box of SOCS1 and SOCS3 mediates the interaction with elongins B and C to connect them with the ubiquitin transferase system [155]. Therefore, SOCS proteins can act as E3 ubiquitin ligases and mediate the degradation of their target proteins such as IRS1 (Figure 2) [156]. Phosphorylation of the SOCS box leads to the degradation of SOCS protein itself, which is necessary to resensitize the IL-6 signaling pathway for acute signaling [157]. 

A disturbance in the balance of inflammatory effectors and their inhibitors can have severe consequences. Mice lacking SOCS1 die before weaning due to dysregulated IFNγ signaling, which causes severe systemic inflammation [148,158]. Conventional deletion of SOCS3 leads to lethality within 11–13 days of gestation as a result of placental defects [159,160]. In HCC cell lines and HCC patient samples, SOCS1 and SOCS3 expression is reduced, caused by aberrant methylation of CpG island in their loci [139,161,162]. Moreover, hepatic SOCS3 deletion results in enhanced chemical-induced fibrosis due to elevated STAT3 activation, which is accompanied with increased TGF-β1 promoting fibrosis [161]. Besides beneficial effects, chronic activation of STAT3 in obesity leads to constant expression of its own inhibitor, SOCS3, disrupting the homeostasis of the IL-6 signaling pathway. Elevated SOCS1 and SOCS3 protein levels impair insulin action by mediating the ubiquitination of IRS1 and its degradation. This in turn contributes to the development of obesity-associated insulin resistance in peripheral organs [163]. In consistency, we identified increased basal activation of STAT3 in hepatocytes from obese mice causing a diminished responsiveness to acute IL-6 [94]. Thus, elevated hepatic SOCS3 levels in obesity impair both insulin and IL-6 signaling, thereby inducing a vicious cycle that might ultimately lead to cancer. 

Collectively, obesity-associated increases of systemic TNFα and IL-6 might impact HCC initiation and progression via redundant and opposing functions of downstream signaling molecules. Therefore, further detailed analyses of signaling capacities in obesity and HCC are required to develop novel approaches to combat this fatal disease.

## 7. Obesity-Induced Inflammation and CRC

Besides HCC, cancer entities of the colon and rectum, colorectal cancer (CRC), also display an elevated burden in obese patients [37]. CRC is the third most common diagnosed cancer type in males and in females, accounting for 9% and 8% of all cancer-related deaths, respectively [164]. Up to 95% of newly diagnosed CRC patients do not carry genetic predispositions, but share common risk factors including age, inflammatory bowel diseases (IBD), and an unhealthy Western lifestyle [165]. In IBD patients, the risk to develop colitis-associated colorectal cancer (CAC) is increased up to 20%, indicating a correlation between the degree of colonic inflammation and CAC risk [166]. A Western lifestyle, characterized by low physical activity, high caloric intake, and excessive smoking, as well as alcohol consumption, favors the development of CRC. Epidemiological studies established a strong correlation between obesity and incidence, as well as morbidity of CRC [36,37,167]. Remarkably, all risk factors—age, IBDs, Western lifestyle—elevate the inflammatory tone not only systemically but also locally in the colon. This inflammation can not only enable but also promote tumor development by causing genomic instability and mutations [168]. 

The capacity of obesity to induce spontaneous CRC development has been supported by genetic mouse models of CRC development. In the commonly used adenomatosis polyposis coli (APC) model for spontaneous and inherited CRC development, obese *APC^Min/+^* mice developed significantly higher numbers of colorectal carcinomas than lean *APC^Min/+^* mice [169,170]. Furthermore, activated K-rat sarcoma (K-ras) signaling led to increased tumor burden in obese *Kras^G12Dint^* mice, due to a shift in microbiota composition [171]. This was accompanied by increased invasiveness and metastasizing capacities. 

Long-term, high-fat diet-feeding alone is already sufficient to induce spontaneous CRC development in C57BL/6 mice [172,173]. Additionally, diet-induced obesity elevates the tumor burden after chemical induction of CAC [174]. Gulhane et al., showed that diet-induced obesity was accompanied with disruption of the intestinal barrier function by reduced expression of mucus-encoding genes and decreased expression of genes encoding tight junctional proteins [175]. Moreover, Ahmad et al., reported an expression switch of *Claudin* genes, integral components of the tight junctions, that resulted in increased permeability of the colonic epithelium in high-fat diet-fed C57BL/6 mice [176]. In the colon, the cell junctional network generates a physical barrier against invading commensal bacteria [177]. Impairment of the intestinal barrier function due to decreased expression of tight junctional proteins results in increased paracellular permeability and translocation of commensal bacteria into the underlying lamina propia, where they evoke an inflammatory response and elevated expression of proinflammatory cytokines [174]. Specifically, diet-induced obesity increases the expression of *Tnfα* in the colons of mice, most likely as a direct consequence of intestinal barrier defects. In line, obese patients exhibit higher circulating endotoxin levels together with higher circulating proinflammatory cytokines, like TNFα and IL-6 [178]. Increased levels of circulating proinflammatory cytokines can ultimately drive CRC development and progression. Data from patients suffering from IBD, such as colitis ulcerosa and Crohn‘s disease, support the link between inflammation and CRC/CAC development via increased circulating TNFα and IL-6 levels [179,180]. Of note, Izano et al., failed to find a significant association between CRC and increased IL-6 and TNFα levels [181]. 

Supporting the importance of increased TNFα and IL-6 in CRC/CAC, TNFR1-, IL-6-, IL-6Rα-deficient mice have reduced mucosal damage, infiltration of macrophages and neutrophils, and consequently less tumors [97,182,183]. This indicates that TNFα and IL-6 signaling promote carcinogenesis and CRC progression in the colon. In contrast, genetic ablation of *Tnf* in *APC^Min/+^* mice exposed to dextran sulfate sodium in the drinking water does not lead to less tumor development [184]. Although, under acute colitis conditions induced by dextran sulfate sodium, TNF-deficient mice and mice treated with anti-TNF exhibit enhanced intestinal inflammation [185], whereas, in mice with chronic inflammation, anti-TNF treatment reduces colitis [186].

However, caloric restriction or reduction of TNFα and IL-6 leads to reduced tumor development in humans and mice by reducing colorectal inflammation [187,188]. In line, recent CAC studies in mice demonstrate that enriching high-fat diets with TNF- and IL-6-inhibiting supplements leads to the reduction of inflammation and mucosal injury in the colons of mice and ultimately to less tumor development [189,190,191,192,193]. Similar effects on tumor burden are seen by reducing obesity-induced inflammation via natural substances in the diets of obese *APC^Min/+^* mice [194,195]. 

Increased TNFα levels are detected in the microenvironment of CRC [35] and vice versa in mouse models for CAC development; TNFα levels are linked with increased leukocyte infiltration and tumor formation [182]. Inactivation of TNFR1 results in reduced tumor burden and transplantation of TNF-Rp55-deficent bone marrow into wild type recipients leads to less tumor development in the CAC model [182]. This indicates that it is not the tumor cells which depend on TNFα signaling, but rather immune cells derived from the bone marrow. In this line, depletion of IKKβ in myeloid cells reduces tumor size and inflammation in CAC [196]. TNFα and TNFR1 are mainly expressed by infiltrating immune cells, but not in the epithelial cells, supporting that TNFα activates NF-κB in inflammatory cells causing colonic inflammation and thereby promoting carcinogenesis. 

Interestingly, the effects of obesity promoting cancer can even be transmitted to the next generation. Maternal high-fat diet consumption enhances offspring susceptibility to dextran sulfate sodium-induced colitis in mice accompanied with upregulated NF-κB signaling, enhanced neutrophil infiltration, and elevated IL-6 levels [197], thereby further revealing TNFα and IL-6 as ideal targets for CRC therapies [198]. 

In contrast to TNFα, IL-6 is a TNFα/ NF-κB-regulated cytokine that acts on epithelial and immune cells. IL-6 is increased in the serum of CRC patients and impacts tumor grade and survival rate [199,200,201,202]. Moreover, IL-6 promotes CAC development partly via its action on intestinal epithelial cells [97,203,204,205,206]. Consistently, a higher number of pSTAT3-expressing epithelial cells are found in CAC patients [207]. Inactivation of IL-6 or the IL-6Rα in mice reduces tumor formation in the CAC mouse model [97,183]. However, mice deficient in intestinal IL-6Rα have a similar tumor load compared to control mice [183,208]. This indicates that IL-6 is not the major signaling pathway in the intestinal epithelium promoting intrinsic tumor growth via STAT3. Rather, IL-6 acts on immune cells in the tumor microenvironment to promote inflammation and tumor growth [183]. Blunting of IL-6 signaling in macrophages reduces tumor development in the CAC mouse model [97,183]. We and others have shown that IL-6Rα-deficient macrophages stay in the proinflammatory M1 state [32,183], while obesity-induced IL-6 shifts macrophage polarization towards the tumor-promoting M2 state. In turn, these M2 macrophages secrete the chemokine CC-chemokine ligand-20 (CCL-20) [183]. In detail, CCL-20 recruits CC-chemokine receptor-6 (CCR-6)-expressing immune cells via chemotaxis into the tumor microenvironment, further driving colonic inflammation and CAC development. 

Taken together, these studies demonstrate that TNFα and IL-6, either induced by obesity or the colonic tumor microenvironment, exert their function mainly on infiltrating immune cells and, to a lesser extent, on intestinal cells to promote CRC. Therefore, future studies will be required to pinpoint exact signaling pathways in obesity and cancer that are amenable for cancer therapy.

## 8. Conclusions

The obesity epidemic is expected to still rise during the next years, and the prevalence of obesity-associated cancer, as well as its treatment expenses, will increase, representing enormous socioeconomic efforts. Given that inflammatory pathways can exert redundant or opposing functions depending on context and cell type, future studies will be required to define exact signaling pathways that contribute to obesity and cancer and are amenable for anticancer therapies. 

## Figures and Tables

**Figure 1 cancers-11-00024-f001:**
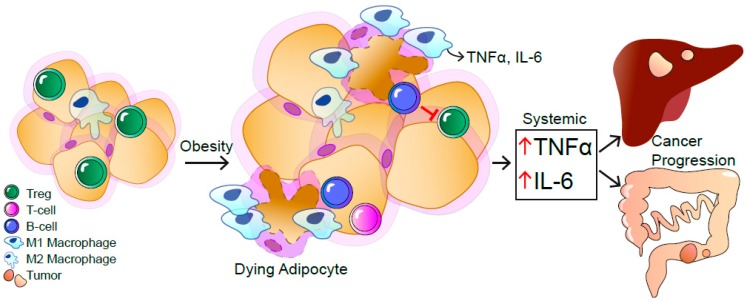
The obesity-induced, systemic, low-grade inflammation contributes to the progression of hepatocellular carcinoma (HCC) and colorectal cancer (CRC). As a response to mechanical and endoplasmatic reticulum stress in obesity, immune cells are recruited to the white adipose tissue (WAT). Dying adipocytes are surrounded by macrophages visible as crown-like structures. In obese WAT, the number of M1-polarized macrophages is increased that release inflammatory cytokines like tumor necrosis factor α (TNFα) and Interleukin-6 (IL-6), resulting in a local and systemic low-grade inflammation. B and T cells are recruited to the obese WAT. Infiltrating B cells inhibit regulatory T cells, further contributing to systemic inflammation accompanied with elevated IL-6 and TNFα levels, causing HCC and CRC.

**Figure 2 cancers-11-00024-f002:**
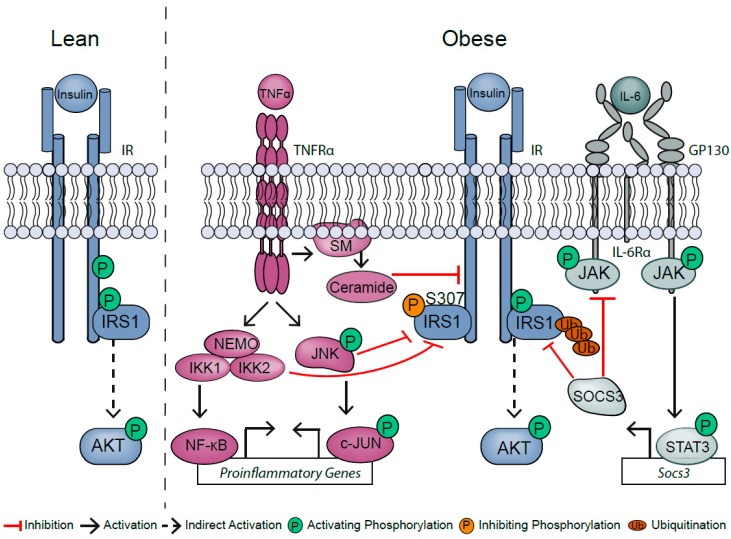
Obesity-induced inflammation causes insulin resistance.

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
