# Peer review of "Obesity-Induced TNFα and IL-6 Signaling: The Missing Link between Obesity and Inflammation—Driven Liver and Colorectal Cancers"

_cancers, 2018, doi:10.3390/cancers11010024_

Round 1

Reviewer 1 Report

Comments to the Authors

The manuscript entitled “Obesity-Induced Inflammation-The Missing Link between Obesity and Cancer” (Manuscript #: cancers-399890) seems to be an interesting manuscript. There are a few points which need to be addressed.

1.      On pages 5-7, lines 172-247 should be divided into several paragraphs.

2.      Lines 350-, authors mention TNF receptor and IL-6. What about the effects of IL-6 receptor or TNF-alpha on CRC development? Any previous reports using KO mouse models?

3.      Authors describe only HCC and CRC. Any reason for choosing these two? Readers may be interested in other types of cancer. Or authors can change the title of this manuscript.

Author Response

The manuscript entitled “Obesity-Induced Inflammation-The Missing Link between Obesity and Cancer” (Manuscript #: cancers-399890) seems to be an interesting manuscript. There are a few points which need to be addressed.

We thank the reviewer for the evaluation of our work.

1.      On pages 5-7, lines 172-247 should be divided into several paragraphs.

We have divided this section to several paragraphs as suggested.

2.      Lines 350-, authors mention TNF receptor and IL-6. What about the effects of IL-6 receptor or TNF-alpha on CRC development? Any previous reports using KO mouse models?

We have mentioned now TNFR, IL-6 and IL-6Ra inactivation in CRC development, but TNF ligand KO has not been investigated yet to our knowledge.

3.      Authors describe only HCC and CRC. Any reason for choosing these two? Readers may be interested in other types of cancer. Or authors can change the title of this manuscript.

We thank the reviewer for pointing towards this direction and have changed the title accordingly.

Reviewer 2 Report

The manuscript is to review literature on the link between obesity and cancer. However, the authors focus on only TNFa and IL-6, too narrow to provide readers a comprehensive view of the link.

Major issues:

Line 21-51, description of the immune cells lacks much details about the number or percentage of cells; only conclusive statements are cited, but do not highlight some key data from the cited references.

Line 53-54, more cytokines and chemokines should be included, such as IL-17.

Line 86-161, how TNFa acts through NF-kB to affect HCC is not discussed in details.

Line 304-387, not much new information is provided and discussed.

Minor issues:

Line 24, the articles that "cannot be cited.." should be cited here to discuss why they do not support the view.

Line 27, give full name for each first use of abbreviations such as "ER".

Line 32-33, what are the differences in ref. 5-6?

Line 37, use "alternative than pro-inflammatory macrophages" is not accurate. The definition of M1 and M2 macrophages should be stated here and used.

Line 55-56, language error "contribution..have, ... reveal".

Line 57-58, how IKK and JNK kinases inhibit IR phosphorylation is not discussed or shown in Figure 1. All figures should have legends. Figure 2 does not make much sense - it is not clear about the source of systemic cytokines: do they all come from adipose tissues or do they also come from circulatory immune cells?

Author Response

reviewer 2

The manuscript is to review literature on the link between obesity and cancer. However, the authors focus on only TNFa and IL-6, too narrow to provide readers a comprehensive view of the link.

We have changed the title to indicate the topic of our review.

Major issues:

Line 21-51, description of the immune cells lacks much details about the number or percentage of cells; only conclusive statements are cited, but do not highlight some key data from the cited references.

We have supplemented details about numbers or percentages for immune cell subsets. Furthermore, we have highlighted more key data from the citations, when necessary. 

Line 53-54, more cytokines and chemokines should be included, such as IL-17.

Since we have changed the focus of our review to TNF and IL-6 in HCC and CRC in obesity-induced inflammation, we have mentioned IL-17 marginally.

Line 86-161, how TNFa acts through NF-kB to affect HCC is not discussed in details.

We have now clearly stated that TNF-induced NFkB affects inflammatory and survival genes to contribute to HCC development.

Line 304-387, not much new information is provided and discussed.

We regret that the reviewer does not find this section useful. We have provided this since we are convinced that negative feedback regulation of IL-6 signaling plays an important role in HCC. There is not much literature out on this topic, but nevertheless understanding of the complete signaling mechanisms might be useful also for readers to interpret the current literature.

Minor issues:

Line 24, the articles that "cannot be cited.." should be cited here to discuss why they do not support the view.

We apologize for this misphrasing and have changed the sentence accordingly.

Line 27, give full name for each first use of abbreviations such as "ER".

We have provided now full names and used abbreviations when applicable.

Line 32-33, what are the differences in ref. 5-6?

Ref 5 described innate immune cells whereas ref 6 evaluated adaptive immune cells in wat inflammation.

Line 37, use "alternative than pro-inflammatory macrophages" is not accurate. The definition of M1 and M2 macrophages should be stated here and used.

We have now provided the nomenclature of Mantovani et al throughout the complete section.

Line 55-56, language error "contribution..have, ... reveal".

Thanks for the carful editing, we have corrected that error.

Line 57-58, how IKK and JNK kinases inhibit IR phosphorylation is not discussed or shown in Figure 1. All figures should have legends. Figure 2 does not make much sense - it is not clear about the source of systemic cytokines: do they all come from adipose tissues or do they also come from circulatory immune cells?

We thank the reviewer for these comments. We have described and cited in the revised version how TNF-induced spingomyelinase and ceramides inhibit IR kinase activity and highlighted this also in changed figure 1. Furthermore, we have revised a new altered Figure 2 that makes more sense.

Reviewer 3 Report

This study is very excellent and shows very well all the mechanisms that underlie inflammation-induced obesity. The inflammation in turn triggers a series of events that in the long run lead to the stimulation of the carcinogenesis processes, encouraging tumor growth and inhibiting the mediators of apoptosis. The role played by the proinflammatory cytokines IL-6 and TNF-alpha is very important. Given  that obesity increases systemic and local levels of these cytokines and TNFalfa and IL-6 are high in  serum of cancer patients, it is not surprising that these cytokines have been associated with an  increased incidence of inflammatory cancer entities. Gathering all the information reported by the authors it is desirable that obesity represents a strong risk factor for those related tumors. So today it is becoming increasingly important to suggest to the population to lead a healthy lifestyle both in terms of their diet and avoiding sedentary life. This leads to an improvement of all metabolism and to extinguishing inflammation by promoting the anti-inflammatory action of factors such as adiponectin in improving insulin sensitivity that is counteracted by TNF-alpha.

Author Response

Reviewer 3

This study is very excellent and shows very well all the mechanisms that underlie inflammation-induced obesity. The inflammation in turn triggers a series of events that in the long run lead to the stimulation of the carcinogenesis processes, encouraging tumor growth and inhibiting the mediators of apoptosis. The role played by the proinflammatory cytokines IL-6 and TNF-alpha is very important. Given  that obesity increases systemic and local levels of these cytokines and TNFalfa and IL-6 are high in  serum of cancer patients, it is not surprising that these cytokines have been associated with an  increased incidence of inflammatory cancer entities. Gathering all the information reported by the authors it is desirable that obesity represents a strong risk factor for those related tumors. So today it is becoming increasingly important to suggest to the population to lead a healthy lifestyle both in terms of their diet and avoiding sedentary life. This leads to an improvement of all metabolism and to extinguishing inflammation by promoting the anti-inflammatory action of factors such as adiponectin in improving insulin sensitivity that is counteracted by TNF-alpha.

We thank the reviewer for the evaluation.

Reviewer 4 Report

This is a literature review of the role of TNF-alpha and IL-6 in hepatocellular and colorectal cancer. Therefore the title is too broad and somewhat misleading. While the review is extensive, there is little synthesis of the information into a review that adds depth to the current literature. This review belongs in a more specialized journal.

Author Response

Reviewer 4

This is a literature review of the role of TNF-alpha and IL-6 in hepatocellular and colorectal cancer. Therefore the title is too broad and somewhat misleading. While the review is extensive, there is little synthesis of the information into a review that adds depth to the current literature. This review belongs in a more specialized journal.

We are regret the reviewer's criticism, but still believe that the reviewed literature are important contributions in the field of obesity-induced TNF and IL-6 in the development of HCC and CRC. Therefore, we have changed the title of the manuscript into a less broader version. While both diverse scientific fields (metabolism and cancer) are covered by this review, it is a tightrope walk to satisfy the broad readership of Cancers in an unilateral direction. Thus, we hope that the reviewer appreciates our efforts in editing the revised version of our manuscript.

Round 2

Reviewer 1 Report

>> Lines 350-, authors mention TNF receptor and IL-6. What about the effects of IL-6 receptor or TNF-alpha on CRC development? Any previous reports using KO mouse models?

> We have mentioned now TNFR, IL-6 and IL-6Ra inactivation in CRC development, but TNF ligand KO has not been investigated yet to our knowledge.

Authors should cite the previous reports below and discuss.

Enhanced intestinal inflammation induced by dextran sulfate sodium in tumor necrosis factor-alpha deficient mice. J Gastroenterol Hepatol. 2003 May;18(5):560-9.

Genetic ablation of Tnfalpha demonstrates no detectable suppressive effect on inflammation-related mouse colon tumorigenesis. Chem Biol Interact. 2010 Mar 30;184(3):423-30.

Author Response

reviewer 1

>> Lines 350-, authors mention TNF receptor and IL-6. What about the effects of IL-6 receptor or TNF-alpha on CRC development? Any previous reports using KO mouse models?

> We have mentioned now TNFR, IL-6 and IL-6Ra inactivation in CRC development, but TNF ligand KO has not been investigated yet to our knowledge.

Authors should cite the previous reports below and discuss.

Enhanced intestinal inflammation induced by dextran sulfate sodium in tumor necrosis factor-alpha deficient mice. J Gastroenterol Hepatol. 2003 May;18(5):560-9.

Genetic ablation of Tnfalpha demonstrates no detectable suppressive effect on inflammation-related mouse colon tumorigenesis. Chem Biol Interact. 2010 Mar 30;184(3):423-30.

we thank the reviewer for his comment and the two interesting manuscripts that we have now described and cited.

Reviewer 2 Report

Responded well.

Author Response

reviewer 2

Responded well.

we thank the reviewer for this comment.

Reviewer 4 Report

Changes are acceptable

Author Response

 reviewer 4

Changes are acceptable

we are grateful to the reviewer.